# Perturbation Learning Based Anomaly Detection

**Jinyu Cai**[1,2]*, **Jicong Fan**[2,3]†
[1]Fuzhou University, Fuzhou, China
[2]Shenzhen Research Institute of Big Data, Shenzhen, China
[3]The Chinese University of Hong Kong, Shenzhen, China
jinyucai1995@gmail.com  fanjicong@cuhk.edu.cn

## Abstract

This paper presents a simple yet effective method for anomaly detection. The main idea is to learn small perturbations to perturb normal data and learn a classifier to classify the normal data and the perturbed data into two different classes. The perturbator and classifier are jointly learned using deep neural networks. Importantly, the perturbations should be as small as possible but the classifier is still able to recognize the perturbed data from unperturbed data. Therefore, the perturbed data are regarded as abnormal data and the classifier provides a decision boundary between the normal data and abnormal data, although the training data do not include any abnormal data. Compared with the state-of-the-art of anomaly detection, our method does not require any assumption about the shape (e.g. hypersphere) of the decision boundary and has fewer hyper-parameters to determine. Empirical studies on benchmark datasets verify the effectiveness and superiority of our method.

## 1 Introduction

Anomaly detection (AD) is an important research problem in many areas such as computer vision, machine learning, and chemical engineering [Chandola *et al.*, 2009; Ramachandra *et al.*, 2020; Pang *et al.*, 2021; Ruff *et al.*, 2021]. AD aims to identify abnormal data from normal data and is usually an unsupervised learning task because the anomaly samples are unknown in the training stage. In the past decades, numerous AD methods [Schölkopf *et al.*, 1999, 2001; Breunig *et al.*, 2000; Liu *et al.*, 2008] have been proposed. For instance, one-class support vector machine (OCSVM) [Schölkopf *et al.*, 2001] maps the data into high-dimensional feature space induced by kernels and tries to find a hyperplane giving the possibly maximal distance between the normal data and the origin. Tax and Duin [2004] proposed a method called support vector data description (SVDD), which finds the smallest hyper-sphere encasing the normal data in a high-dimensional feature space. SVDD is similar to OCSVM with a Gaussian kernel function.

Classical anomaly detection methods such as OCSVM and SVDD are generally not suitable for large-scale data due to the high computational costs and are not effective to deal with more complex data such as those in vision scenarios. To address these issues, a few researchers [Erfani *et al.*, 2016; Golan and El-Yaniv, 2018; Abati *et al.*, 2019; Wang *et al.*, 2019; Qiu *et al.*, 2021] attempted to take advantage of deep learning [LeCun *et al.*, 2015; Goodfellow *et al.*, 2016] to improve the performance of anomaly detection. One typical way is to use a deep auto-encoder or its variants[Vincent *et al.*, 2008; Kingma and Welling, 2013; Pidhorskyi *et al.*, 2018; Wang *et al.*, 2021] to learn effective data representation or compression models. Auto-encoder and its variants have achieved promising performance in AD. In fact, these methods do not explicitly define an objective for anomaly detection. Instead, they usually use the data reconstruction error as a metric to detect anomalies.

---

*The work was done during the visiting at SRIBD and CUHK-Shenzhen.
†Corresponding author

36th Conference on Neural Information Processing Systems (NeurIPS 2022).

There are a few approaches to building deep anomaly detection objectives or models. Typical examples include classical one-class learning based approach[Ruff *et al.*, 2018, 2019; Perera and Patel, 2019; Bhattacharya *et al.*, 2021], probability estimation based approach [Zong *et al.*, 2018; Pérez-Cabo *et al.*, 2019; Su *et al.*, 2019], and adversarial learning based approach [Deecke *et al.*, 2018; Perera *et al.*, 2019; Raghuram *et al.*, 2021]. For example, Ruff *et al.* [2018] proposed deep one-class classification (DSVDD), which applies deep neural network to learn an effective embedding for the normal data such that in the embedding space, the normal data can be encased by a hyper-sphere with minimum radius. The deep autoencoding Gaussian mixture model (DAGMM) proposed by Zong *et al.* [2018] is composed of a compression network and an estimation network based on the Gaussian mixture model. The anomaly score is defined as the output energy of the estimation network. Perera *et al.* [2019] proposed one-class GAN (OCGAN) to learn a latent space to represent a specific class by adversarially training the auto-encoder and discriminator. The properly trained OCGAN network can well reconstruct the specific class of data, while failing to reconstruct other classes of data. Moreover, some latest works also explore interesting perspectives. Goyal *et al.* [2020] proposed the method called deep robust one-class classification (DROCC). DROCC assumes that the normal samples generally lie on low-dimensional manifolds, and regards the process of finding the optimal hyper-sphere in the embedding space as an adversarial optimization problem. Yan *et al.* [2021] claimed that anomalous domains generally exhibit different semantic patterns compared with the peripheral domains, and proposed the semantic context based anomaly detection network (SCADN) to learn the semantic context from the masked data via adversarial learning. Chen *et al.* [2022] proposed the interpolated Gaussian descriptor (IGD) to learn more valid data descriptions from representative normal samples rather than edge samples. Shenkar and Wolf [2022] utilized contrastive learning to construct the method called generic one-class classification (GOCC) for AD on tabular data. It is worth noting that classical AD methods such as OCSVM [Schölkopf *et al.*, 2001] and DSVDD [Ruff *et al.*, 2018] require specific assumptions (e.g. hypersphere) for the distribution or structure of the normal data. The GAN-based approaches [Deecke *et al.*, 2018; Perera *et al.*, 2019] suffer from the instability problem of min-max optimization and have high computational costs.

In this paper, we propose a novel AD method called perturbation learning based anomaly detection (PLAD). PLAD aims to learn a perturbator and a classifier from the normal training data. The perturbator uses minimum effort to perturb the normal data to abnormal data while the classifier is able to classify the normal data and perturbed data into two classes correctly. The main contributions of our work are summarized as follows:

- We propose a novel AD method called PLAD. PLAD does not require any assumption about the shape of the decision boundary between the normal data and abnormal data. In addition, PLAD has much fewer hyper-parameters than many state-of-the-art AD methods such as [Wang *et al.*, 2019; Goyal *et al.*, 2020; Yan *et al.*, 2021].

- We propose to learn perturbations directly from the normal training data. For every training data point, we learn a distribution from which any sample can lead to a perturbation such that the normal data point is flipped to an abnormal data point.

- Besides the conventional empirical studies on one-class classification, we investigate the performance of our PLAD and its competitors in recognizing abnormal data from multi-class normal data. These results show that our PLAD has state-of-the-art performance.

- Although our experiments are mainly on images and tabular data, PLAD is actually a framework of AD and can also be applied to time series, text, and graph data via changing the network components.

## 2 Proposed method

Suppose we have a distribution $\mathcal{D}$ of dimension $d$ and any data drawn from $\mathcal{D}$ are deemed as normal data. Now we have some training data $\mathbb{X} = \{\mathbf{x}_1, \mathbf{x}_2, \ldots, \mathbf{x}_n\}$ randomly drawn from $\mathcal{D}$ and we want to learn a discriminative function $f$ from $\mathbb{X}$ such that $f(\mathbf{x}) > 0$ for any $\mathbf{x} \in \mathcal{D}$ and $f(\mathbf{x}) < 0$ for any $\mathbf{x} \notin \mathcal{D}$. This is an unsupervised learning problem and also known as anomaly detection, where any $\mathbf{x} \notin \mathcal{D}$ are deemed as abnormal data.

In contrast to classical anomaly detection methods such as one-class SVM [Schölkopf *et al.*, 2001], deep SVDD [Ruff *et al.*, 2018] , and DROCC [Goyal *et al.*, 2020], in this paper, we do not make any

assumption about the distribution $\mathcal{D}$. Inspired from the generative adversarial learning (GAN) [Goodfellow *et al.*, 2014], we propose to learn a discriminator that is able to recognize fake data (abnormal data) and a generator that is able to fool the discriminator, but the input of the generator is real data (normal data), which is essentially different from GAN. To be more precise, we propose to learn perturbations for $\mathbf{x}$ such that the perturbed $\widetilde{\mathbb{X}}$ (denoted by $\widetilde{\mathbb{X}} = \{\tilde{\mathbf{x}}_1, \tilde{\mathbf{x}}_2, \ldots, \tilde{\mathbf{x}}_n\}$) are abnormal but quite close to $\mathbb{X}$. To ensure the abnormality of $\widetilde{\mathbb{X}}$, we learn a classifier $f$ from $\{\mathbb{X}, \widetilde{\mathbb{X}}\}$ such that $f(\mathbf{x}) > 0$ for any $\mathbf{x} \in \mathbb{X}$ and $f(\tilde{\mathbf{x}}) < 0$ for any $\tilde{\mathbf{x}} \in \widetilde{\mathbb{X}}$. To ensure that $\widetilde{\mathbb{X}}$ is close to $\mathbb{X}$, the perturbations should be small enough. Therefore, we propose to solve the following problem

$$\underset{\theta, \widetilde{\mathbb{X}}}{\text{minimize}} \ \frac{1}{n} \sum_{i=1}^{n} \ell(y_i, f_\theta(\mathbf{x}_i)) + \frac{1}{n} \sum_{i=1}^{n} \ell(\tilde{y}_i, f_\theta(\tilde{\mathbf{x}}_i)) + \frac{\lambda}{n} \sum_{i=1}^{n} \phi(\mathbf{x}_i, \tilde{\mathbf{x}}_i), \tag{1}$$

where $\ell(\cdot, \cdot)$ denotes some loss function such as cross-entropy and $y_1 = \cdots = y_n = 0$ and $\tilde{y}_1 = \cdots = \tilde{y}_n = 1$ are the labels for the normal data and perturbed data respectively. $\theta$ denotes the set of parameters of the classifier $f$ and $\phi(\cdot, \cdot)$ is some distance metric quantifying the difference between two data points. $\lambda$ is a hyperparameter to control the magnitudes of the perturbations. However, directly solving (1) encounters the following difficulties.

- First, the number $(|\theta| + dn)$ of decision variables to optimize can be huge if $n$ is large, where $|\theta|$ denotes the cardinality of the set $\theta$.

- Second, it is hard to use mini-batch optimization because some decision variables (i.e. $\widetilde{\mathbb{X}}$) are associated with the sample indices.

- Lastly, it is not easy to determine $\phi$ because $\phi$ relies on the unknown distribution $\mathcal{D}$. For instance, $\phi(\mathbf{x}, \tilde{\mathbf{x}}) = \|\mathbf{x} - \tilde{\mathbf{x}}\|^2$, namely the squared Euclidean norm, does not work if $\mathbb{X}$ is enclosed by a hypersphere or hypercube (data points close to the centroid require much larger perturbations than those far away from the centroid, which implies that $\mathbf{x} - \tilde{\mathbf{x}}$ has a non-Gaussian distribution).

To overcome these three difficulties, we propose to solve the following problem instead

$$\underset{\theta, \tilde{\theta}}{\text{minimize}} \ \frac{1}{n} \sum_{i=1}^{n} \ell(y_i, f_\theta(\mathbf{x}_i)) + \frac{1}{n} \sum_{i=1}^{n} \ell(\tilde{y}_i, f_\theta(\tilde{\mathbf{x}}_i)) + \frac{\lambda}{n} \sum_{i=1}^{n} \left( \|\boldsymbol{\alpha}_i - \mathbf{1}\|^2 + \|\boldsymbol{\beta}_i - \mathbf{0}\|^2 \right)$$

$$\text{subject to} \ \tilde{\mathbf{x}}_i = \mathbf{x}_i \odot \boldsymbol{\alpha}_i + \boldsymbol{\beta}_i, \ (\boldsymbol{\alpha}_i, \boldsymbol{\beta}_i) = g_{\tilde{\theta}}(\mathbf{x}_i), \ i = 1, 2, \ldots, n, \tag{2}$$

where $\mathbf{1} = [1, 1, \ldots, 1]^\top$ and $\mathbf{0} = [0, 0, \ldots, 0]^\top$ are $d$-dimensional constant vectors and $\odot$ denotes the Hadamard product of two vectors. $\boldsymbol{\alpha}_i$ and $\boldsymbol{\beta}_i$ are multiplicative and additive perturbations for $\mathbf{x}_i$ and they are generated from a perturbator $g_{\tilde{\theta}}$, where $\tilde{\theta}$ denotes the set of parameters to learn. In (2), we hope that the multiplicative perturbation is close to 1 and the additive perturbation is close to 0 but they rely on the data point $\mathbf{x}$. Therefore, the perturbation learning is adaptive to the unknown distribution $\mathcal{D}$, which solves the third aforementioned difficulty.

In fact, problem (2) can be reformulated as

$$\underset{\theta, \tilde{\theta}}{\text{minimize}} \ \frac{1}{n} \sum_{i=1}^{n} \left( \ell\big(y_i, f_\theta(\mathbf{x}_i)\big) + \ell\big(\tilde{y}_i, f_\theta(\mathbf{x}_i \odot g_{\tilde{\theta}}^{\alpha}(\mathbf{x}_i) + g_{\tilde{\theta}}^{\beta}(\mathbf{x}_i))\big) \right)$$

$$+ \frac{\lambda}{n} \sum_{i=1}^{n} \left( \|g_{\tilde{\theta}}^{\alpha}(\mathbf{x}_i) - \mathbf{1}\|^2 + \|g_{\tilde{\theta}}^{\beta}(\mathbf{x}_i) - \mathbf{0}\|^2 \right), \tag{3}$$

where $\begin{bmatrix} g_{\tilde{\theta}}^{\alpha}(\mathbf{x}_i) \\ g_{\tilde{\theta}}^{\beta}(\mathbf{x}_i) \end{bmatrix} = g_{\tilde{\theta}}(\mathbf{x}_i)$, $i = 1, 2, \ldots, n$. We see that we only need to optimize the parameters $\theta$ and $\tilde{\theta}$ and the total number of decision variables is $|\theta| + |\tilde{\theta}|$, which solved the first difficulty we discussed previously. On the other hand, $\theta$ and $\tilde{\theta}$ are not associated with the sample indices, which solved the second difficulty. Once $f_\theta$ and $g_{\tilde{\theta}}$ are learned, we can then use $f_\theta$ to detect whether a new data point $\mathbf{x}_{\text{new}}$ is normal (e.g. $f_\theta(\mathbf{x}_{\text{new}}) < 0.5$) or abnormal (e.g. $f_\theta(\mathbf{x}_{\text{new}}) > 0.5$). We call the method *Perturbation Learning based Anomaly Detection* (PLAD).

In PLAD, namely (3), both the classifier $f_\theta$ and the perturbator $g_{\tilde{\theta}}$ are neural networks. They can be fully connected neural networks, convolutional neural networks (for image data anomaly detection),

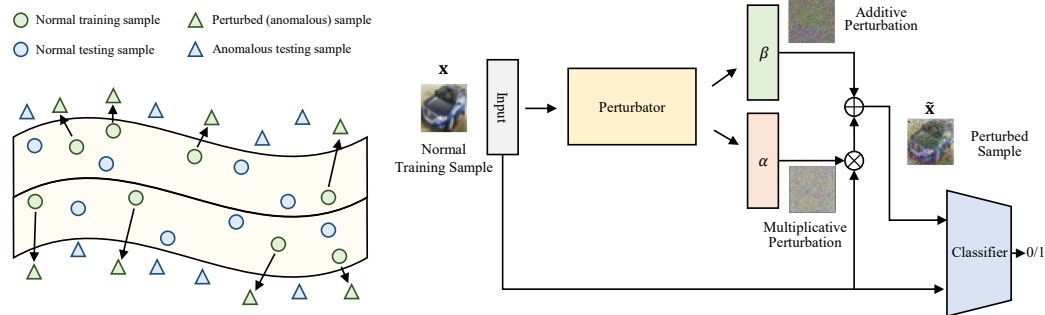

Figure 1: The motivation and network architecture of the proposed PLAD method.

or recurrent neural networks (for sequential data anomaly detection). Figure 1 shows the motivation of our PLAD and the network architecture. Compared with many popular anomaly detection methods, our PLAD has the following characteristics.

- PLAD does not make any assumption about the distribution or structure of the normal data. In contrast, one-class SVM [Schölkopf *et al.*, 2001], deep SVDD [Ruff *et al.*, 2018], and DROC-C [Goyal *et al.*, 2020] make specific assumptions about the distribution of the normal data, which may be violated in real applications. For example, deep SVDD assumes that the normal data are encased by a hypersphere, which is hard to guarantee in real applications and may require a very deep neural network to transform a non-hypersphere structure to a hypersphere structure.

- In PLAD, besides the network structures, we only need to determine one hyperparameter $\lambda$, which provides huge convenience in real applications. In contrast, many state-of-the-art AD methods such as [Wang *et al.*, 2019; Goyal *et al.*, 2020; Yan *et al.*, 2021] have at least two key hyperparameters. For example, in DROCC [Goyal *et al.*, 2020], one has to determine the radius of the hyper-sphere, the step size of gradient-ascent, and two regularization parameters.

- In PLAD, we can use gradient-based optimizer such as Adam to solve the optimization. The time complexity is comparable to that of vanilla deep neural networks (for classification or representation). On the contrary, many recent advances DROCC [Goyal *et al.*, 2020] of anomaly detection especially those GAN based methods [Deecke *et al.*, 2018; Perera *et al.*, 2019; Yan *et al.*, 2021] have much higher computational costs.

As shown in the left plot of Figure 1, normal training data can be perturbed to abnormal data by different perturbations. Therefore, we propose to learn a distribution from which any perturbations can perturb the normal training data to be abnormal. Specifically, for every $\mathbf{x}_i$, there exists a distribution $\Omega_i$, such that for any $\mathbf{z} \in \Omega_i$, the perturbation given by $(\boldsymbol{\alpha}, \boldsymbol{\beta}) = h(\mathbf{z})$ can perturb $\mathbf{x}_i$ to be abnormal, where $h$ is a nonlinear function modelled by a neural network. We can just assume that $\Omega_i$ is a Gaussian distribution with mean $\boldsymbol{\mu}_i$ and variance $\boldsymbol{\sigma}_i^2$, i.e., $\Omega_i = \mathcal{N}(\boldsymbol{\mu}_i, \boldsymbol{\sigma}_i^2)$, because of the universal approximation ability of $h$. We take the idea of variational autoencoder (VAE) [Kingma and Welling, 2013] and minimize

$$\mathcal{L}(\tilde{\theta}_1, \tilde{\theta}_2) = D_{KL}\left(q_{\tilde{\theta}_1}(\mathbf{z}|\mathbf{x})\|p(\mathbf{z})\right) - \mathbb{E}_{q_{\tilde{\theta}_2}(\mathbf{z}|\mathbf{x})}\left[\log p_{\tilde{\theta}_2}(\boldsymbol{\varepsilon}|\mathbf{z})\right], \tag{4}$$

where $p(\mathbf{z}) = \mathcal{N}(\mathbf{0}, \mathbf{1})$, $\boldsymbol{\varepsilon} = \begin{bmatrix} \mathbf{1} \\ \mathbf{0} \end{bmatrix}$, $\mathbf{z} = l_{\tilde{\theta}_1}(\mathbf{x})$, $(\boldsymbol{\alpha}, \boldsymbol{\beta}) = h_{\tilde{\theta}_2}(\mathbf{z})$, $\tilde{\theta}_1$ are the parameters of the encoder $l$, and $\tilde{\theta}_2$ are the parameters of the decoder $h$. Combing (4) with (3), we have $\tilde{\theta} = \{\tilde{\theta}_1, \tilde{\theta}_2\}$ and solve

$$\begin{aligned} \operatorname*{minimize}_{\theta, \tilde{\theta}} \quad & \frac{1}{n}\sum_{i=1}^{n}\left(\ell\big(y_i, f_\theta(\mathbf{x}_i)\big) + \ell\big(\tilde{y}_i, f_\theta(\mathbf{x}_i \odot g_{\tilde{\theta}}^{\alpha}(\mathbf{x}_i) + g_{\tilde{\theta}}^{\beta}(\mathbf{x}_i))\big)\right) \\ & + \frac{1}{n}D_{KL}\left(q_{\tilde{\theta}_1}(\mathbf{z}|\mathbf{x})\|p(\mathbf{z})\right) - \lambda\mathbb{E}_{q_{\tilde{\theta}_2}(\mathbf{z}|\mathbf{x})}\left[\log p_{\tilde{\theta}_2}(\boldsymbol{\varepsilon}|\mathbf{z})\right], \end{aligned} \tag{5}$$

where $g_{\tilde{\theta}} = h_{\tilde{\theta}_2} \circ l_{\tilde{\theta}_1}$. The training is similar to that for VAE [Kingma and Welling, 2013] and will not be detailed here.

It should be pointed out that the method (5) is just an extension of the method (3). In (5), we want to learn a distribution for each normal training data point $\mathbf{x}_i$ such that any perturbations generated from the distribution can perturb $\mathbf{x}_i$ to be abnormal. It is expected that (5) can outperform (3) in real applications. The corresponding experiments are in Appendix D.

# 3   Connection with previous works

The well-known one-class classification methods such as OCSVM [Schölkopf *et al.*, 2001], DSVD-D [Ruff *et al.*, 2018] and DROCC [Goyal *et al.*, 2020] have specific assumptions for the embedded distribution while our PLAD does not require any assumption and is able to adaptively learn a decision boundary even if it is very complex. It is also noteworthy that the idea of DROCC in identifying anomalies is slightly similar to ours, i.e, training a classifier instead of an auto-encoder or embedding model.

Self-supervised learning based methods, e.g., GOAD [Bergman and Hoshen, 2020] NeuTraL AD [Qiu *et al.*, 2021], etc., work through designing an appropriate auxiliary task to help the model to learn useful data features. Both our approach and the self-supervised learning based methods aim to learn a binary classifier on the original data and auxiliary data. But they are different in the following two points. First, in self-supervised learning (e.g. contrastive learning) based methods, the auxiliary data are generated by some pre-defined and manual operations (e.g. image rotation and cropping). In contrast, in our approach, the auxiliary data are adaptively and dynamically learned from the original data. Second, self-supervised learning based methods are often designed for some specific types of data such as image data, whereas it may be difficult to generate auxiliary data for other types of data. In contrast, it is easy to apply our approach to any type of data.

Adversarial learning based methods [Malhotra *et al.*, 2016; Deecke *et al.*, 2018; Pidhorskyi *et al.*, 2018; Perera *et al.*, 2019] are generally constructed with auto-encoder and GAN [Goodfellow *et al.*, 2014], and the most widely used measure of them to detect anomalies is the reconstruction error. Compared with them, we exploit the idea of VAE when producing perturbations and the detection decision is made by a classifier, which should be more suitable than reconstruction error for anomaly detection. On the other hand, in these adversarial learning based AD methods, the min-max optimization leads to instabilities in detecting anomalies, while the optimization of PLAD is much easier to solve. Another interesting work SCADN [Yan *et al.*, 2021] tries to produce negative samples by multi-scale striped masks to train a GAN, but its anomaly score still relies on reconstruction error and the production of masks has randomness or may be hard to determine in various real scenarios. Our PLAD learns perturbations adaptively from the data itself, which is convenient and reliable.

# 4   Experiment

In this section, we evaluate the proposed method in comparison to several state-of-the-art anomaly detection methods on two image datasets and two tabular datasets. Note that all the compared methods do not utilize any pre-trained feature extractors.

## 4.1   Datasets and baseline methods

**Datasets description**

- **CIFAR-10:** CIFAR-10 image dataset is composed of 60,000 images in total, where 50,000 samples for training and 10,000 samples for test. It includes 10 different balanced classes.
- **Fashion-MNIST:** Fashion MNIST contains 10 different categories of grey-scale fashion style objects. The data is split into 60,000 images for training and 10,000 images for test.
- **Thyroid:** Thyroid is a hypothyroid disease dataset that contains 3,772 samples with 3 classes and 6 attributes. We follow the data split settings of [Zong *et al.*, 2018] to preprocess the data for one-class classification task.
- **Arrhythmia:** Arrhythmia dataset consists of 452 samples with 274 attributes. Here we also follow the data split settings of [Zong *et al.*, 2018] to preprocess the data.

The detailed information of each dataset is illustrated in Table 1.

Table 1: Details of the datasets used in our experiments.

| Dataset name | Type | # Total samples | # Dimension |
|---|---|---|---|
| CIFAR-10 | Image | 60,000 | $32 \times 32 \times 3$ |
| Fashion-MNIST | Image | 70,000 | $28 \times 28$ |
| Thyroid | Tabular | 3,772 | 6 |
| Arrhythmia | Tabular | 452 | 274 |

**Baselines and state-of-the-arts.** We compare our method with the following classical baseline methods and state-of-the-art methods: OCSVM [Schölkopf *et al.*, 2001], isolation forest (IF) [Liu *et al.*, 2008], local outlier factor (LOF) [Breunig *et al.*, 2000], denoising auto-encoder (DAE) [Vincent *et al.*, 2008], E2E-AE and DAGMM [Zong *et al.*, 2018], DCN [Caron *et al.*, 2018], ADGAN [Deecke *et al.*, 2018], DSVDD [Ruff *et al.*, 2018], OCGAN [Perera *et al.*, 2019], TQM [Wang *et al.*, 2019], GOAD [Bergman and Hoshen, 2020], DROCC [Goyal *et al.*, 2020], HRN-L2 and HRN [Hu *et al.*, 2020], SCADN [Yan *et al.*, 2021], IGD (Scratch) [Chen *et al.*, 2022], NeuTraL AD [Qiu *et al.*, 2021], and GOCC [Shenkar and Wolf, 2022].

## 4.2 Implementation details and evaluation metrics

In this section, we first describe the implementation details of the proposed PLAD method. The settings for image and tabular datasets are illustrated as follows:

- **Image datasets.** For image datasets (CIFAR-10 and Fashion-MNIST), we utilize the LeNet-based CNN to construct the classifier, which is same as [Ruff *et al.*, 2018] and [Goyal *et al.*, 2020] to provide fair comparison. And we apply the MLP-based VAE to learn the noise for data. Since both image datasets contain 10 different classes, it can be regarded as 10 independent one-class classification tasks, and each task on CIFAR-10 has 5,000 training samples (6,000 for Fashion-MNIST) and 10,000 testing samples for both of them. Consequently, the choice of optimizer (from Adam [Kingma and Ba, 2015] and SGD), learning rate and hyper-parameter $\lambda$ could be varies for different classes. The suggested settings of them on each experiment in this paper refer to Appendix A.

- **Tabular datasets.** For tabular datasets (Thyroid and Arrhythmia), we both use the MLP-based classifier and VAE in practice, and we train them by Adam optimizer with learning rate 0.001. Besides, $\lambda$ is set to 3 for Thyroid and 2 for Arrhythmia.

For the compared methods in the experiment, we report their performance directly from the following paper [Hu *et al.*, 2020; Goyal *et al.*, 2020; Yan *et al.*, 2021; Qiu *et al.*, 2021; Chen *et al.*, 2022; Shenkar and Wolf, 2022] except for DROCC, which we run the official released code to obtain the results. Due to the limitation of paper length, the details of the our network settings are provided in Appendix A.

Regarding the evaluation metrics, we follow the previous works such as [Ruff *et al.*, 2018] and [Zong *et al.*, 2018] to use AUC (Area Under the ROC curve) for image datasets and F1-score for tabular datasets because the anomaly detection for image and tabular datasets has different evaluation criteria. Moreover, our method does not need pre-training like [Ruff *et al.*, 2018] and others did, so we train the proposed method 5 times with 100 epochs to obtain the average performance and standard deviation. Note that we run all experiments on NVIDIA RTX3080 GPU with 32GB RAM, CUDA 11.0 and cuDNN 8.0.

## 4.3 Experimental results on image datasets

Table 2 and Table 3 summarize the average AUCs performance of the one-class classification tasks on CIFAR-10 and Fashion-MNIST, where we have the following observations:

- Compared with some classical shallow model-based approaches such OCSVM and IF, the proposed PLAD method significantly outperforms them on each one-class classification task with a large margin. This is mainly due to the powerful feature learning capability of deep neural network.

Table 2: Average AUCs (%) in one-class anomaly detection on CIFAR-10. Note that for the compared methods we only report their mean performance, while we further report the standard deviation for the proposed method. * denotes we run the officially released code to obtain the results. In each case, the best two results are marked in **bold**.

| Normal Class | Airplane | Auto mobile | Bird | Cat | Deer | Dog | Frog | Horse | Ship | Truck |
|---|---|---|---|---|---|---|---|---|---|---|
| OCSVM [Schölkopf *et al.*, 2001] | 61.6 | 63.8 | 50.0 | 55.9 | 66.0 | 62.4 | 74.7 | 62.6 | 74.9 | 75.9 |
| IF [Liu *et al.*, 2008] | 66.1 | 43.7 | 64.3 | 50.5 | 74.3 | 52.3 | 70.7 | 53.0 | 69.1 | 53.2 |
| DAE[Vincent *et al.*, 2008] | 41.1 | 47.8 | 61.6 | 56.2 | 72.8 | 51.3 | 68.8 | 49.7 | 48.7 | 37.8 |
| DAGMM [Zong *et al.*, 2018] | 41.4 | 57.1 | 53.8 | 51.2 | 52.2 | 49.3 | 64.9 | 55.3 | 51.9 | 54.2 |
| ADGAN [Deecke *et al.*, 2018] | 63.2 | 52.9 | 58.0 | 60.6 | 60.7 | 65.9 | 61.1 | 63.0 | 74.4 | 64.2 |
| DSVDD [Ruff *et al.*, 2018] | 61.7 | 65.9 | 50.8 | 59.1 | 60.9 | 65.7 | 67.7 | 67.3 | 75.9 | 73.1 |
| OCGAN [Perera *et al.*, 2019] | 75.7 | 53.1 | 64.0 | 62.0 | 72.3 | 62.0 | 72.3 | 57.5 | 82.0 | 55.4 |
| TQM [Wang *et al.*, 2019] | 40.7 | 53.1 | 41.7 | 58.2 | 39.2 | 62.6 | 55.1 | 63.1 | 48.6 | 58.7 |
| DROCC* [Goyal *et al.*, 2020] | 79.2 | **74.9** | **68.3** | 62.3 | 70.3 | 66.1 | 68.1 | **71.3** | 62.3 | **76.6** |
| HRN-L2 [Hu *et al.*, 2020] | **80.6** | 48.2 | 64.9 | 57.4 | **73.3** | 61.0 | 74.1 | 55.5 | 79.9 | 71.6 |
| HRN [Hu *et al.*, 2020] | 77.3 | 69.9 | 60.6 | **64.4** | 71.5 | **67.4** | **77.4** | 64.9 | **82.5** | 77.3 |
| PLAD | **82.5** (0.4) | **80.8** (0.9) | 68.8 (1.2) | 65.2 (1.2) | 71.6 (1.1) | 71.2 (1.6) | 76.4 (1.9) | 73.5 (1.0) | 80.6 (1.8) | 80.5 (1.3) |

Table 3: Average AUCs (%) and in one-class anomaly detection on Fashion-MNIST. Note that for the compared methods we only report their mean performance, while we further report the standard deviation for the proposed method. * denotes we run the officially released code to obtain the results. In each case, the best two results are marked in **bold**.

| Normal Class | T-shirt | Trouser | Pullover | Dress | Coat | Sandal | Shirt | Sneaker | Bag | Ankle boot |
|---|---|---|---|---|---|---|---|---|---|---|
| OCSVM [Schölkopf *et al.*, 2001] | 86.1 | 93.9 | 85.6 | 85.9 | 84.6 | 81.3 | 78.6 | 97.6 | 79.5 | 97.8 |
| IF [Liu *et al.*, 2008] | 91.0 | 97.8 | 87.2 | 93.2 | 90.5 | 93.0 | 80.2 | 98.2 | 88.7 | 95.4 |
| DAE[Vincent *et al.*, 2008] | 86.7 | 97.8 | 80.8 | 91.4 | 86.5 | 92.1 | 73.8 | 97.7 | 78.2 | 96.3 |
| DAGMM [Zong *et al.*, 2018] | 42.1 | 55.1 | 50.4 | 57.0 | 26.9 | 70.5 | 48.3 | 83.5 | 49.9 | 34.0 |
| ADGAN [Deecke *et al.*, 2018] | 89.9 | 81.9 | 87.6 | 91.2 | 86.5 | 89.6 | 74.3 | 97.2 | 89.0 | 97.1 |
| DSVDD [Ruff *et al.*, 2018] | 79.1 | 94.0 | 83.0 | 82.9 | 87.0 | 80.3 | 74.9 | 94.2 | 79.1 | 93.2 |
| OCGAN [Perera *et al.*, 2019] | 85.5 | 93.4 | 85.0 | 88.1 | 85.8 | 88.5 | 77.5 | 93.9 | 82.7 | 97.8 |
| TQM [Wang *et al.*, 2019] | 92.2 | 95.8 | **89.9** | 93.0 | **92.2** | 89.4 | **84.4** | 98.0 | **94.5** | 98.3 |
| DROCC* [Goyal *et al.*, 2020] | 88.1 | 97.7 | 87.6 | 87.7 | 87.2 | 91.0 | 77.1 | 95.3 | 82.7 | 95.9 |
| HRN-L2 [Hu *et al.*, 2020] | 91.5 | 97.6 | 88.2 | 92.7 | 91.0 | 71.9 | 79.4 | **98.9** | 90.8 | **98.9** |
| HRN [Hu *et al.*, 2020] | **92.7** | 98.5 | 88.5 | 93.1 | 92.1 | **91.3** | 79.8 | **99.0** | **94.6** | 98.8 |
| PLAD | **93.1** (0.5) | **98.6** (0.2) | 90.2 (0.7) | 93.7 (0.6) | 92.8 (0.8) | **96.0** (0.4) | 82.0 (0.6) | 98.6 (0.3) | 90.9 (1.0) | **99.1** (0.1) |

- PLAD also explicitly outperforms several well-known deep anomaly detection methods such as DAGMM and DSVDD, and consistently obtains the top two AUC scores on most classes of CIFAR-10 and Fashion-MNIST compared to some latest methods such as TQM, DROCC and HRN. Specifically, on the class "Automobile" of CIFAR-10 and class "Sandal" of Fashion-MNIST, the AUC improvements of the proposed method exceed 5.9% and 4.7% respectively compared to the runner-up. We also conduct statistical analysis for our method against other competitors, the detailed analysis refers to Section 4.6.

- It is noteworthy that some deep anomaly detection methods such as DSVDD and DROCC, are mainly based on the assumption that the normal data in the embedding space are situated in a hyper-sphere, while anomalies are outside the sphere. Therefore the edge of the hyper-sphere is then the decision boundary learned by the model to identify anomalies. In contrast, PLAD does not require any assumption about the shape of the decision boundary. It attempts to learn the perturbation from data itself by neural network and construct the anomalies by enforcing the perturbation to original data, then train the network to distinguish the normal samples and anomalies. This is natural to relate PLAD with some adversarial learning based methods such as ADGAN and OCGAN, etc. In contrast to them, the optimization of PLAD is a non-adversary problem and hence is easier to solve.

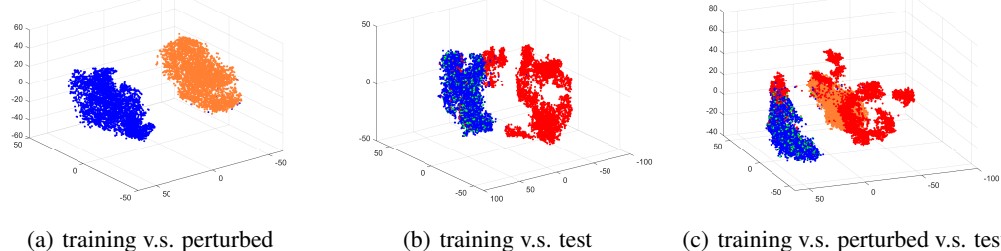

| (a) training v.s. perturbed | (b) training v.s. test | (c) training v.s. perturbed v.s. test |

Figure 2: The learned embedding space visualized by t-SNE. Note the points marked in blue, orange, green, and red are training samples, perturbed samples, normal test samples, and anomalous test samples, respectively. We select the class of "Sneaker" in Fashion-MNIST to provide the visualization. More results for the other categories can be found in Appendix B.

Table 4 shows the average performance on CIFAR-10 and Fashion-MNIST over all 10 classes to provide an overall comparison. Note that two more latest methods SCADN and IGD (Scratch) are also compared in the table, though their performance on each single class were not reported in their papers. From the table we observe that PLAD achieves the best average AUCs on both datasets among all compared methods.

Table 4: Average AUCs (%) over all 10 classes on CIFAR-10 and Fashion-MNIST. Note that the best two results are marked in **bold**.

| Data set | CIFAR-10 | Fashion-MNIST |
|---|---|---|
| OCSVM [Schölkopf *et al.*, 2001] | 64.7 | 87.0 |
| IF [Liu *et al.*, 2008] | 59.7 | 91.5 |
| DAE[Vincent *et al.*, 2008] | 53.5 | 88.1 |
| DAGMM [Zong *et al.*, 2018] | 53.1 | 51.7 |
| ADGAN [Deecke *et al.*, 2018] | 62.4 | 88.4 |
| DSVDD [Ruff *et al.*, 2018] | 64.8 | 84.7 |
| OCGAN [Perera *et al.*, 2019] | 65.6 | 87.8 |
| TQM [Wang *et al.*, 2019] | 52.1 | 92.7 |
| DROCC* [Goyal *et al.*, 2020] | 69.9 | 89.0 |
| HRN-L2 [Hu *et al.*, 2020] | 66.6 | 90.0 |
| HRN [Hu *et al.*, 2020] | 71.3 | **92.8** |
| SCADN [Yan *et al.*, 2021] | 66.9 | — |
| IGD (Scratch) [Chen *et al.*, 2022] | **74.3** | 92.0 |
| PLAD | **75.1** | **93.5** |

We use t-SNE [Van der Maaten and Hinton, 2008] to visualize the learned embedding features of PLAD. Specifically, we visualize the training samples of a specific category, perturbed data for training samples, as well as all test samples. They are divided into three sub-figures for ease of understanding. Figure 2 shows the visualization of the class "Sneaker" in Fashion-MNIST. From this figure we have the following observations. First, PLAD obviously learns good decision boundary for distinguishing normal data and perturbed data from Figure 2(a). Second, Figures 2(b) and 2(c) show that the normal test samples lie in the same manifold as the training samples, while the abnormal samples (abnormal test samples and perturbed samples) are relatively separated. In other words, PLAD explicitly learns a discriminative embedding space to distinguish normal samples and anomalies. More results can be found in Appendix B.

## 4.4 Experimental results on non-image datasets

Table 5 summarizes the F1-scores of each compared method on the Thyroid and Arrhythmia datasets. It can be observed that PLAD significantly outperforms several baseline methods such as OCSVM, DAGMM, DSVDD and DROCC with a large margin. Although NeuTraL AD and GOCC achieve encouraging 76.8% F1-score on Thyroid, it is worth mentioning that they are both methods designed for non-image data. The Arrhythmia dataset seems to be a more difficult anomaly detection task

because of its small sample size, which is not conducive to deep learning. Surprisingly, the proposed PLAD method shows remarkable performance on Arrhythmia, which surpasses 9.2% compared to the runner-up. Moreover, the performance of PLAD is also comparable to NeuTraL AD and GOCC on Thyroid, which fully demonstrates its applicability to the anomaly detection task for non-image data.

Table 5: Average F1-scores (%) with the standard deviation of each method on the two tabular datasets (Thyroid and Arrhythmia). * denotes we run the official released code to obtain the results, and the best two results are marked in **bold**.

| Data set | Thyroid | Arrhythmia |
|---|---|---|
| OCSVM [Schölkopf *et al.*, 2001] | $39.0 \pm 1.0$ | $46.0 \pm 0.0$ |
| LOF [Breunig *et al.*, 2000] | $54.0 \pm 1.0$ | $51.0 \pm 1.0$ |
| E2E-AE[Zong *et al.*, 2018] | $13.0 \pm 4.0$ | $45.0 \pm 3.0$ |
| DCN [Caron *et al.*, 2018] | $33.0 \pm 3.0$ | $38.0 \pm 3.0$ |
| DAGMM [Zong *et al.*, 2018] | $49.0 \pm 4.0$ | $49.0 \pm 3.0$ |
| DSVDD [Ruff *et al.*, 2018] | $73.0 \pm 0.0$ | $54.0 \pm 1.0$ |
| DROCC* [Goyal *et al.*, 2020] | $68.7 \pm 2.3$ | $32.3 \pm 1.8$ |
| GOAD [Bergman and Hoshen, 2020] | $74.5 \pm 1.1$ | $52.0 \pm 2.3$ |
| NeuTraL AD [Qiu *et al.*, 2021] | $\mathbf{76.8 \pm 1.9}$ | $60.3 \pm 1.1$ |
| GOCC [Shenkar and Wolf, 2022] | $\mathbf{76.8 \pm 1.2}$ | $\mathbf{61.8 \pm 1.8}$ |
| PLAD | $76.6 \pm 0.6$ | $\mathbf{71.0 \pm 1.7}$ |

## 4.5 Experiment of separating anomaly from multi-class normal data

It should be pointed out that in real applications, the normal data may contain multiple classes without labels. We need to separate anomalies from these multi-class normal data. In this study, we randomly select 10,000 samples among the training split of CIFAR-10 or Fashion-MNIST to construct a new normal training set, namely, the normal data in multiple classes. Subsequently, we randomly select two samples from the test split of CIFAR-10 or Fashion-MNIST to construct 10,000 anomalous samples using the means of pair-wise samples at the pixel level. The produced anomalous samples are merged with the original test split to form a new test set. Compared with the previous two tasks (Sections 4.3 and 4.4), this one is much more difficult because the decision boundary between the anomalous samples and the normal samples is very complicated. We run the experiment to compare our method with four baselines including OCSVM, DAGMM, DSVDD, DROCC and report the results in Table 6. We see that the proposed PLAD method outperforms the baselines significantly. For example, the improvement over the runner-up DSVDD are 9.1% and 4.4% on CIFAR-10 and Fashion-MNIST respectively. The success of PLAD mainly stems from the ability of learning a decision boundary adaptively without any assumption.

Table 6: AUCs (%) of the experiment of separating anomaly from multi-class normal data.

| Data set | CIFAR-10 | Fashion-MNIST |
|---|---|---|
| OCSVM [Schölkopf *et al.*, 2001] | $54.9 \pm 0.0$ | $64.8 \pm 0.0$ |
| DAGMM [Zong *et al.*, 2018] | $44.3 \pm 0.6$ | $49.2 \pm 2.6$ |
| DSVDD [Ruff *et al.*, 2018] | $63.6 \pm 1.1$ | $70.9 \pm 2.0$ |
| DROCC [Goyal *et al.*, 2020] | $60.9 \pm 5.8$ | $68.1 \pm 3.1$ |
| PLAD | $\mathbf{72.7 \pm 1.9}$ | $\mathbf{75.3 \pm 2.8}$ |

## 4.6 Statistical analysis of the proposed method

To assess whether the results obtained of our method are statistically significantly different compared to others, we further conduct statistical analysis for them. Student's t-test is a common scheme for assessing the significance of differences between two groups of data. Usually, the difference is said to be significant if the $p$-value obtained in the t-test is less than 0.05. We compare with three baseline methods including DSVDD [Ruff *et al.*, 2018], DROCC [Goyal *et al.*, 2020], and HRN [Hu *et al.*, 2020] on Fashion-MNIST. Note that we perform t-test experimental by using the results obtained from running their official released codes. Table 7 summarizes the reproduced results and the $p$-value

(t-test) of our method against other baseline methods. We can see that the reproduced results are close to that reported in the related papers [Hu *et al.*, 2020; Goyal *et al.*, 2020], and the t-test results of our methods are statistically significantly different ($p < 0.01$) from other comparative methods in most cases except for the comparison with HRN on the "Trouser" and "Dress" classes.

Table 7: The reproduced results (with *), reported results, and t-test results of the one-class anomaly detection task on Fashion-MNIST.

| Reproduce | T-shirt | Trouser | Pullover | Dress | Coat | Sandal | Shirt | Sneaker | Bag | Ankle boot |
|---|---|---|---|---|---|---|---|---|---|---|
| DSVDD [Ruff *et al.*, 2018] | 79.1 | 94.0 | 83.0 | 82.9 | 87.0 | 80.3 | 74.9 | 94.2 | 79.1 | 93.2 |
| DSVDD* [Ruff *et al.*, 2018] | $78.4 \pm 3.3$ | $93.6 \pm 1.3$ | $80.8 \pm 3.4$ | $84.1 \pm 2.0$ | $85.9 \pm 2.4$ | $82.0 \pm 3.0$ | $75.0 \pm 3.8$ | $94.5 \pm 1.8$ | $80.6 \pm 5.9$ | $94.1 \pm 1.5$ |
| DROCC* [Goyal *et al.*, 2020] | $88.1 \pm 3.3$ | $97.7 \pm 0.7$ | $87.6 \pm 1.4$ | $87.7 \pm 1.6$ | $87.2 \pm 2.2$ | $91.0 \pm 1.6$ | $77.1 \pm 2.0$ | $95.3 \pm 0.7$ | $82.7 \pm 2.9$ | $95.9 \pm 2.1$ |
| HRN [Hu *et al.*, 2020] | $92.7 \pm 0.0$ | $98.5 \pm 0.1$ | $88.5 \pm 0.1$ | $93.1 \pm 0.1$ | $92.1 \pm 0.1$ | $91.3 \pm 0.4$ | $79.8 \pm 0.1$ | $99.0 \pm 0.0$ | $94.6 \pm 0.1$ | $98.8 \pm 0.0$ |
| HRN* [Hu *et al.*, 2020] | $88.8 \pm 0.1$ | $98.6 \pm 0.1$ | $84.8 \pm 0.1$ | $93.2 \pm 0.1$ | $89.5 \pm 0.2$ | $89.6 \pm 0.1$ | $74.4 \pm 0.1$ | $98.9 \pm 0.0$ | $87.2 \pm 0.3$ | $97.7 \pm 0.1$ |
| PLAD (Ours) | $93.1 \pm 0.5$ | $98.6 \pm 0.2$ | $90.2 \pm 0.7$ | $93.7 \pm 0.6$ | $92.8 \pm 0.8$ | $96.0 \pm 0.4$ | $82.0 \pm 0.6$ | $98.6 \pm 0.3$ | $90.9 \pm 1.0$ | $99.1 \pm 0.1$ |
| *p*-value (t-test) | T-shirt | Trouser | Pullover | Dress | Coat | Sandal | Shirt | Sneaker | Bag | Ankle boot |
| v.s. DSVDD | $1.5 \times 10^{-7}$ | $1.2 \times 10^{-6}$ | $2.2 \times 10^{-5}$ | $1.0 \times 10^{-7}$ | $2.1 \times 10^{-5}$ | $8.2 \times 10^{-8}$ | $8.7 \times 10^{-4}$ | $4.7 \times 10^{-5}$ | 0.003 | $2.9 \times 10^{-6}$ |
| v.s. DROCC | $9.4 \times 10^{-4}$ | 0.004 | $3.3 \times 10^{-4}$ | $3.4 \times 10^{-6}$ | $3.0 \times 10^{-5}$ | $3.1 \times 10^{-6}$ | $4.4 \times 10^{-4}$ | $3.4 \times 10^{-7}$ | $9.9 \times 10^{-6}$ | $5.1 \times 10^{-4}$ |
| v.s. HRN | $4.0 \times 10^{-9}$ | 0.614 | $1.2 \times 10^{-7}$ | 0.013 | $5.9 \times 10^{-6}$ | $3.0 \times 10^{-10}$ | $1.5 \times 10^{-9}$ | $5.4 \times 10^{-4}$ | $2.2 \times 10^{-5}$ | $1.7 \times 10^{-10}$ |

## 4.7 Extension of PLAD to time-series anomaly detection

We further conduct experiment to demonstrate the effectiveness of PLAD on the time-series anomaly detection task, the details refer to Table 8 in Appendix G due to the limitation of the paper length.

## 5 Conclusion

We have presented a novel method PLAD for anomaly detection. Compared with its competitors, PLAD does not require any assumption about the distribution or structure of the normal data. This is the major reason that PLAD outperforms its competitors. In addition, PLAD has fewer hyperparameters to determine and has a lower computation cost than many strong baselines such as [Wang *et al.*, 2019; Goyal *et al.*, 2020; Yan *et al.*, 2021]. Actually, PLAD provides us a framework for anomaly detection. Different neural networks such as CNN [Krizhevsky *et al.*, 2012], RNN [Mikolov *et al.*, 2010], GNN [Scarselli *et al.*, 2008], and even transformer Vaswani *et al.* [2017] can be embedded into PLAD to accomplish various anomaly detection tasks such as time series anomaly detection. One limitation of our work is that we haven't included enough these experiments currently.

Note that in this study, we only considered the additive and multiplicative perturbations, i.e., $\tilde{\mathbf{x}} = \mathbf{x} \odot g_{\tilde{\theta}}^{\alpha}(\mathbf{x}) + g_{\tilde{\theta}}^{\beta}(\mathbf{x})$, while a more general perturbation can be formulated as $\tilde{\mathbf{x}} = g_{\tilde{\theta}}'(\mathbf{x})$. When we need a very sophisticated perturbator, we may not well approximate $g_{\tilde{\theta}}'(\mathbf{x})$ by $\mathbf{x} \odot g_{\tilde{\theta}}^{\alpha}(\mathbf{x}) + g_{\tilde{\theta}}^{\beta}(\mathbf{x})$, because we have to made some restriction for $g_{\tilde{\theta}}^{\alpha}$ and $g_{\tilde{\theta}}^{\beta}$. Therefore, future study may also focus on determining the $\phi$ for (1), to learn $g_{\tilde{\theta}}'(\mathbf{x})$ more effectively.

## Acknowledgments

The work was supported by the Youth program 62106211 of the National Natural Science Foundation of China and the research funding T00120210002 of Shenzhen Research Institute of Big Data.

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
