# OpenReview forum: "Perturbation Learning Based Anomaly Detection"
_NeurIPS.cc/2022/Conference — NeurIPS 2022 Accept_

### Official Review · Reviewer_ZTAh · 2022-07-09

**Rating:** 7
**Confidence:** 3
**Soundness:** 3 good
**Presentation:** 3 good
**Contribution:** 3 good

**Summary:**

The authors propose an approach to semi-supervised anomaly detection. The core idea is to generate anomalous data using an adaptive perturbator, which modifies the normal data and hence, solve the problem of anomaly detection as a supervised binary classification task.

**Questions:**

What impact does the perturbation operation have on the inferred normal data distribution boundaries?
How is the approach related and how does it compare to self-supervision based anomaly detection methods?

**Limitations:**

It is not clear how the learned class of the perturbed samples can capture samples which can't be obtained by a small perturbation of the normal data class.
The approach hasn't been tested on time-series anomaly detection as also stated by the authors.

**Strengths And Weaknesses:**

The idea is original and the results are promising. It would have been helpful to have the presented approach compared to self-supervised approaches to anomaly detection as these seem well related to the perturbation idea.
The perturbation to the original data has been chosen to be either additive or multiplicative, it is not discussed what are the limitations on the inferred boundaries of the normal data distribution given only these 2 perturbations.

---

> ### Author Response · Authors · 2022-08-02
> **Thanks for the review. We added the t-SNE visualization of the training data, perturbed data, and testing data, the influence of $\lambda$, and the experiment of time-series anomaly detection.**
>
>
> **Response to questions:**
>
> **Q1: What impact does the perturbation operation have on the inferred normal data distribution boundaries?**
>
> **Response:** We thank the reviewer for the insightful question. To show the impact of the perturbation operation on the inferred normal data distribution boundaries, we provided more visualization results in the main paper and supplementary material. Specifically, in Figure 2 (Page 8) of the main paper, we use t-SNE to visualize the features (extracted by the classifier) of training (normal) data and perturbed data, normal testing data, and abnormal testing data. We can observe that PLAD obviously learns a good separation between the training (normal) data and perturbed data. The separation is also useful for the normal testing data and abnormal testing data.  Moreover, the normal testing data are close to the training data, while the abnormal testing data are close to the perturbed data. This demonstrates that PLAD learns an effective discriminator to distinguish normal data and abnormal data. It is worth mentioning that the 3-D visualization is just for an intuitive understanding of the impact of the perturbation operation and the decision boundary of PLAD in higher-dimensional space should be more meaningful.
>
> Besides, we added some results about the influence of regularization parameter $\lambda$ on the detection performance, which can be found in Figure 4 of the supplementary material. Theoretically, a large $\lambda$ corresponds a weak perturbation, which leads a tight decision boundary of PLAD. Generally, according to Figure 4, PLAD is not very sensitive to $\lambda$ and work well when $\lambda$ is not too large (e.g. 100) or too small (e.g. 0.1).
>
> **Q2: How is the approach related and how does it compare to self-supervised based anomaly detection methods? It would have been helpful to have the presented approach compared to self-supervised approaches to anomaly detection as these seem well related to the perturbation idea.**
>
> **Response:** Thanks for raising this question. Both our approach and the self-supervised learning based methods aim to learn a binary classifier on the original data and auxiliary data. But they are different in the following two points.
>
> First, in self-supervised learning (e.g. contrastive learning) based methods, the auxiliary data are generated by some pre-defined and manual operations (e.g. image rotation and cropping). In contrast, in our approach, the auxiliary data are adaptively and dynamically learned from the original data.
>
> Second, self-supervised learning based methods are often designed for some specific types of data such as image data, whereas it may be difficult to generate auxiliary data for other types of data. In contrast, it is easy to apply our approach to any type of data.
>
> We added these discussions in Section 3 of the main paper.
>
> In fact, we have already compared some self-supervised learning based anomaly detection methods, GOAD [Bergman and Hoshen, 2020] and NeuTraLAD [Qiu et al., 2021], on Section 4.3.2. We present the  results in the following table, in which our method has competitive performance compared to them.
>
> $$
> \begin{matrix}
> \textrm{Data set}  & \textrm{Thyroid} & \textrm{Arrhythmia} \\\\
> \hline
> \textrm{GOAD [Bergman and Hoshen, 2020]}   &74.5 \pm 1.1   & 52.0 \pm 2.3  \\\\
> \textrm{NeuTraL AD [Qiu et al., 2021]} &\bf 76.8 \pm 1.9    &60.3 \pm 1.1  \\\\
> \hline
> \textrm{PLAD}  & 76.6 \pm 0.6 &\bf71.0 \pm 1.7 \\\\
> \end{matrix}
> $$
>
> **Q3: The perturbation to the original data has been chosen to be either additive or multiplicative, it is not discussed what are the limitations on the inferred boundaries of the normal data distribution given only these 2 perturbations.**
>
> **Response:** We added the following discussion to the conclusion section.
>
> Note that in this study, we only considered the additive and multiplicative perturbations, i.e.,
> $\tilde{\mathbf{x}}=\mathbf{x} \odot g_{\tilde{\theta}}^{\alpha}\left(\mathbf{x}\right)+g_{\tilde{\theta}}^{\beta}\left(\mathbf{x}\right)$,
> while a more general perturbation can be formulated as $\tilde{\mathbf{x}}=g'\_{\tilde{\theta}}\left(\mathbf{x}\right)$. When we need a very sophisticated perturbator, we may not well approximate $g'\_{\tilde{\theta}}\left(\mathbf{x}\right)$ by $\mathbf{x} \odot g\_{\tilde{\theta}}^{\alpha}\left(\mathbf{x}\right)+g\_{\tilde{\theta}}^{\beta}\left(\mathbf{x}\right)$, because we have to made some restriction for $g\_{\tilde{\theta}}^{\alpha}$ and $g\_{\tilde{\theta}}^{\beta}$. Therefore, future study may also focus on determining the $\phi$ for eq. (1), to learn $g'_{\tilde{\theta}}\left(\mathbf{x}\right)$ more effectively.
>
> Due to the maximum number of words, please find the responses to the limitations below.

---

> > ### Author Response · Authors · 2022-08-02
> > **Thanks for the review. We added the t-SNE visualization of the training data, perturbed data, and testing data, the influence of , and the experiment of time-series anomaly detection.**
> >
> > **Response on limitations:**
> >
> > **L1: It is not clear how the learned class of the perturbed samples can capture samples which can't be obtained by a small perturbation of the normal data class.**
> >
> > **Response:** Actually, the biggest problem is that we do not know how large a perturbation should be. For example, a too large perturbation would make the abnormal samples too easily to be recognized, which will lead to a weak classifier that cannot distinguish between normal and abnormal data. A too small perturbation may lead to the failure of training the classifier. In addition, it is very difficult to know the type of distribution of required perturbation or noise in advance. Therefore, we propose to learn a perturbator adaptively. As shown in Figure 4 in the supplementary material, our PLAD works well within a wide range of the hyperparameter $\lambda$.
> >
> > **L2: The approach hasn't been tested on time-series anomaly detection as also stated by the authors.**
> >
> > **Response:** Thanks for your constructive comment. We have supplemented the experiment (on an Epileptic Seizure dataset) of time-series anomaly detection. In PLAD, we replaced the MLP with LSTM for the time-series data. The results are shown in the following table. Our PLAD outperformed all the baselines. We also conduct statistical analysis for our method against the runner-up DROCC using Student's t-test, the $p$-value of t-test indicates the difference between two sets of data. Usually, the difference is said to be significant if the $p$-value is less than 0.05. Because of the page length limitation, we have to put the experimental setting and results in the supplementary material.
> > $$
> > \begin{matrix}
> > \hline
> > \textrm{Method}  & k\textrm{-NN}    & \textrm{AE [2014]} & \textrm{DAGMM [2018]} & \textrm{DSVDD [2018]} &\textrm{DROCC [2020]} & \textrm{PLAD}   & p\textrm{-value (t-test) v.s DROCC}\\\\
> > \hline
> > \textrm{AUC} \pm \textrm{std} &91.7 \pm 0.0  &91.5 \pm 1.9 &87.0 \pm 1.07 &94.3 \pm 2.1 &98.1 \pm 0.5 &\bf98.6 \pm 0.3 &0.03\\\\
> > \hline
> > \end{matrix}
> > $$
> >
> > As mentioned in the conclusion section, our PLAD is simple yet effective. It may have many interesting variants and applications.
> >
> > **Thank you again for recognizing our contribution.**

---

### Official Review · Reviewer_hfVr · 2022-07-11

**Rating:** 6
**Confidence:** 4
**Soundness:** 3 good
**Presentation:** 3 good
**Contribution:** 3 good

**Summary:**

This paper proposes PLAD, i.e., Perturbation Learning-based Anomaly Detection, that learns to inject noise into the normal samples to produce abnormal samples for anomaly detection. The approach has been evaluated on image data as well as tabular data, achieving promising anomaly detection (AD) results.

**Questions:**

It would be interesting to see how PLAD will perform on sequential data (e.g. multivariate time series anomaly detection), as also recognized by the authors. Including one or two experiments in the aspect will greatly improve the generality and application potentials of the proposed method. Even adding some discussions will help a lot.

**Limitations:**

The authors have properly discussed the limitations of the work in the conclusion.

**Strengths And Weaknesses:**

## Strengths

1. The presentation is clear. The paper offers a clear presentation with a good explanation of the methodology.
2. Comprehensive comparisons and discussions with related works (e.g., Sec. 3).
3. Good empirical evaluations. PLAD is evaluated on both tabular and image datasets and achieves competitive results.

## Weaknesses
1. The structure of the paper could be more compact. From my perspective, the illustrations of the learned patterns of the injected perturbation are particularly insightful and interesting, and it would be better if they (or some of them, if considering the space limit) can appear in the main text.

---

> ### Author Response · Authors · 2022-08-02
> **Thanks for the suggestions. We have added the experiment of time-series anomaly detection.**
>
> **Response to the questions:**
>
> **Q1:The structure of the paper could be more compact. From my perspective, the illustrations of the learned patterns of the injected perturbation are particularly insightful and interesting, and it would be better if they (or some of them, if considering the space limit) can appear in the main text.**
>
> **Response:** Thanks for the suggestion. We have revised the paper and moved some images illustrating the learned patterns to the main paper, which can be found in Figure 2 (Page 8) in the main text.
>
> **Q2: It would be interesting to see how PLAD will perform on sequential data (e.g. multivariate time-series anomaly detection), as also recognized by the authors. Including one or two experiments in the aspect will greatly improve the generality and application potentials of the proposed method. Even adding some discussions will help a lot.**
>
> **Response:** Thanks for your constructive comment. We have supplemented the experiment (on an Epileptic Seizure dataset) of time-series anomaly detection. In PLAD, we replaced the MLP with LSTM for the time-series data. The results are shown in the following table. Our PLAD outperformed all the baselines. We also conduct statistical analysis for our method against the runner-up DROCC using Student's t-test. The 0.03 $p$-value of t-test indicates our method PLAD is significantly better than DROCC. Because of the page length limitation, we have to put the experimental setting and results in the supplementary material.
>
> $$
> \begin{matrix}
> \hline
> \textrm{Method}  & k\textrm{-NN}    & \textrm{AE [2014]} & \textrm{DAGMM [2018]} & \textrm{DSVDD [2018]} &\textrm{DROCC [2020]} & \textrm{PLAD}   & p\textrm{-value (t-test) v.s DROCC}\\\\
> \hline
> \textrm{AUC} \pm \textrm{std} &91.7 \pm 0.0  &91.5 \pm 1.9 &87.0 \pm 1.07 &94.3 \pm 2.1 &98.1 \pm 0.5 &\bf98.6 \pm 0.3 &0.03\\\\
> \hline
> \end{matrix}
> $$
>
> By the way, we believe that there is a big room for improvement. For example, we may insert transformer or GNN into PLAD. We may extend the idea of PLAD to other tasks such as image denoising and missing data imputation. We hope our work can make a significant contribution to the community.
>
> **Thank you again for recognizing our work.**

---

> > ### Comment · Reviewer_hfVr · 2022-08-05
> > **Post rebuttal comment**
> >
> > Thank the authors for the reply and for making revisions to improve the paper. This increases my confidence in leaning toward supporting the paper for publication.

---

### Official Review · Reviewer_dTj9 · 2022-07-12

**Rating:** 4
**Confidence:** 4
**Soundness:** 2 fair
**Presentation:** 2 fair
**Contribution:** 2 fair

**Summary:**

The paper presents PLAD, an anomaly detection methodology that relies on perturbations of the original data with the goal of learning to separate the perturbed data from the original, which helps in the detection of anomalies. An evaluation across multiple datasets and against multiple baseline methodologies provide evidence on the promise of PLAD.

**Questions:**

The idea of constructing perturbations and then learning models to separate them from the original data is interesting with good potential for the AD problem. Unfortunately, there are several concerns with this work:

W1. Lack of technical novelty and depth

There is no proper introduction of relevant concepts in preliminaries. The approach reads as if it's the first work using perturbations and goes ahead and presents how the problem is solved with Equations 2/3.  No motivation, nor how these loss functions were obtained is provided.

In addition, there is a lack of technical depth. What's used in the loss function is more or less existing knowledge. Likely combined together for the first time, but unclear how this advances the area of perturbation-based learning (i.e., the paper reads as an application of such solutions to AD - heavy re-writing is needed in that direction to introduce concepts and prior work that led to PLAD).

W2. Experiments used "as is" from other papers

The experimentation is weak. Even though many baselines are used, results are taken "as is" from previous papers, without detailed evaluation and tuning as the presented approach.

It's not clear if the results are statistically significantly different. A rigorous statistical analysis is desirable.



**Strengths And Weaknesses:**

Strengths:

S1. Perturbation-base learning for AD is a timely and important topic

S2. Experimentation with many baselines and open datasets

S3. Code is provided

Weaknesses:

W1. Lack of technical novelty and depth

W2. Experiments used "as is" from other papers

---

> ### Author Response · Authors · 2022-08-02
> **Thanks for the review. We added more discussion about the motivation and previous work and included more numerical results.**
>
> **Response to the questions:**
>
> **Q1: Lack of technical novelty and depth**
>
> "...no proper introduction of relevant concepts in preliminaries", "No motivation, nor how these loss functions were obtained is provided", "a lack of technical depth", "unclear how this advances the area of perturbation-based learning", "heavy re-writing is needed in that direction to introduce concepts and prior work that led to PLAD".
>
> **Response:** Thanks for raising these concerns. Actually, to the best of our knowledge, this is the first work of learning perturbations for anomaly detection. Therefore, in the introduction, we did not provide enough relevant concepts. Instead, we discussed the limitations of existing anomaly detection methods: "It is worth noting that classical AD methods such as OCSVM [Schölkopf et al., 2001] and DSVDD [Ruff et al., 2018] require specific assumptions (e.g. hypersphere) for the distribution or structure of the normal data. The GAN-based approaches [Deecke et al., 2018; Perera et al., 2019] suffer from the instability problem of min-max optimization and have high computational costs."  Moreover, in Section 3, we discussed the connection of our method to previous work including one-class classification based methods and adversarial learning based methods. Nevertheless, the connections are not that strong, which in return further verified the novelty of our method.
>
> In this revised paper, we have added one paragraph in Section 3 to show the connection of our method to self-supervised learning based methods:
>
> "Self-supervised learning based methods, e.g., GOAD [Bergman and Hoshen, 2020],  NeuTraL AD [Qiu et al., 2021], etc., work through designing an appropriate auxiliary task to help the model to learn useful data features. Both our approach and the self-supervised learning based methods aim to learn a binary classifier on the original data and auxiliary data. But they are different in the following two points. First, in self-supervised learning (e.g. contrastive learning) based methods, the auxiliary data are generated by some pre-defined and manual operations (e.g. image rotation and cropping). In contrast, in our approach, the auxiliary data are adaptively and dynamically learned from the original data. Second, self-supervised learning based methods are often designed for some specific types of data such as image data, whereas it may be difficult to generate auxiliary data for other types of data. In contrast, it is easy to apply our approach to any type of data."
>
> In sum, there is almost no prior work that led to our PLAD. In this revision, we added the following sentences to the second paragraph of Section 2 to show the inspiration we got from adversarial learning [Goodfellow et al., 2014], though our network is very different from GAN (the input of GAN is random noise while the input of our network is the observed data):
>
> "In contrast to classical anomaly detection methods such as one-class SVM [Schölkopf et al., 2001],
> deep SVDD [Ruff et al., 2018], and DROCC [Goyal et al., 2020], in this paper, we do not make any assumption about the distribution $\mathcal{D}$. Inspired from the generative adversarial learning [Goodfellow et al., 2014], we propose to learn a discriminator that is able to recognize fake data (abnormal data) and a generator that is able to fool the discriminator, but the input of the generator is real data (normal data), which is essentially different from GAN. To be more precise, we propose to learn perturbations for $\mathbf{X}$ such that the perturbed $\mathbb{X}$ (denoted by $\widetilde{\mathbb{X}}=\{\tilde{\mathbf{x}}_1,\tilde{\mathbf{x}}_2,\ldots,\tilde{\mathbf{x}}_n\}$) are abnormal but quite close to $\mathbb{X}$. To ensure the abnormality of $\widetilde{\mathbb{X}}$, we learn a classifier $f$ from  $\{\mathbb{X},\widetilde{\mathbb{X}}\}$ such that
> $f(\mathbf{x})>0$ for any $\mathbf{x}\in\mathbb{X}$ and $f(\tilde{\mathbf{x}})<0$ for any $\tilde{\mathbf{x}}\in\widetilde{\mathbb{X}}$. To ensure that $\widetilde{\mathbb{X}}$ is close to $\mathbb{X}$, the perturbations should be small enough..."
>
> Indeed, this is an empirical work that does not include any theorems. Actually, we tried to analyze the generalization ability of the proposed method. However, the training data for the classifier are paired, including an original sample and its perturbed version, not independent. Therefore, this is out of the scope of the data assumptions of existing deep learning theories. We cannot use the i.i.d Rademacher complexity  or the non-i.i.d Rademacher complexity (e.g. $\beta$-mixing processes) to derive the generalization error bound for PLAD. But is it interesting to study it in future.
>
> Please kindly find the responses to Q2 and Q3 given below, due to the limitation of words.

---

> > ### Author Response · Authors · 2022-08-02
> > **We reproduced a few baselines and report the standard deviations and the p-value of t-test to show the statistical significances.**
> >
> > **Q2: Experiments used "as is" from other papers**
> >
> > "Even though many baselines are used, results are taken "as is" from previous papers, without detailed evaluation and tuning as the presented approach"
> >
> > **Response:** Thanks for the comment. First, we have to clarify that the experimental setup (e.g., the data split of training and testing set, the definition of normal and abnormal samples, etc) of our method is exactly the same as those of the competitors such as DAGMM [Zong et al., 2018], DSVDD [Ruff et al., 2018], DROCC [Goyal et al., 2020], HRN [Hu et al., 2020]. Thus we think it is reasonable and fair to use the reported performance of their papers. This is a convention followed by many previous works such as the DROCC [Goyal et al., 2020] and HRN [Hu et al., 2020].
> >
> > Second, in this revision, we added the reproduced results of many latest works such as DSVDD [Ruff et al., 2018], DROCC [Goyal et al., 2020], and HRN [Hu et al., 2020]. We used their released codes and tuned the hyperparameters carefully. The results of them on Fashion-MNIST are shown as follows (* represents the reproduced results). We also include them in the revised version of Table 3 in the main text, as well as in the supplementary material. We can see that the reproduced performances are close to those reported in the related papers. For DROCC, we just ran the released code to get the results since there are no results of Fashion-MNIST in the related papers.
> > $$
> > \begin{matrix}
> > \textrm{Reproduce} & \textrm{T-shirt}    & \textrm{Trouser} & \textrm{Pullover} & \textrm{Dress} & \textrm{Coat} & \textrm{Sandal} & \textrm{Shirt} & \textrm{Sneaker} & \textrm{Bag} & \textrm{Ankle boot} \\\\
> > \hline
> > \textrm{DSVDD}       & 79.1 & 94.0 & 83.0 & 82.9 & 87.0 & 80.3 & 74.9 & 94.2 & 79.1 & 93.2  \\\\
> > \textrm{DSVDD}^*    &78.4 \pm 3.3 &93.6 \pm 1.3 &80.8 \pm 3.4 &84.1 \pm 2.0 &85.9 \pm 2.4 &82.0 \pm 3.0 &75.0 \pm 3.8 &94.5 \pm 1.8 &80.6 \pm 5.9 &94.1 \pm 1.5 \\\\
> > \textrm{DROCC}^*    &88.1 \pm 3.3 &97.7 \pm 0.7 &87.6 \pm 1.4 &87.7 \pm 1.6 &87.2 \pm 2.2 &91.0 \pm 1.6 &77.1 \pm 2.0 &95.3 \pm 0.7 &82.7 \pm 2.9 &95.9 \pm 2.1 \\\\
> > \textrm{HRN}       & 92.7 \pm 0.0 & 98.5 \pm 0.1 & 88.5 \pm 0.1 & 93.1 \pm 0.1 & 92.1 \pm 0.1 & 91.3 \pm 0.4 & 79.8 \pm 0.1 & 99.0 \pm 0.0 & 94.6 \pm 0.1 & 98.8 \pm 0.0 \\\\
> > \textrm{HRN}^*    &88.8 \pm 0.1 &98.6 \pm 0.1 &84.8 \pm 0.1 &93.2 \pm 0.1 &89.5 \pm 0.2 &89.6 \pm 0.1 &74.4 \pm 0.1 &98.9 \pm 0.0 &87.2 \pm 0.3 &97.7 \pm 0.1  \\\\
> > \textrm{PLAD(Ours)}     & 93.1 \pm 0.5 &  98.6 \pm 0.2 & 90.2 \pm 0.7  & 93.7 \pm 0.6   & 92.8 \pm 0.8 &96.0 \pm 0.4 & 82.0 \pm 0.6 & 98.6 \pm 0.3 & 90.9 \pm 1.0  & 99.1 \pm 0.1 \\\\
> > \hline
> > p\textrm{-value (t-test)}& \textrm{T-shirt}    & \textrm{Trouser} & \textrm{Pullover} & \textrm{Dress} & \textrm{Coat} & \textrm{Sandal} & \textrm{Shirt} & \textrm{Sneaker} & \textrm{Bag} & \textrm{Ankle boot}  \\\\
> > \hline
> > \textrm{v.s. DSVDD } &1.5\times10^{-7} &1.2\times10^{-6} &2.2\times10^{-5} &1.0\times10^{-7} & 2.1\times10^{-5} & 8.2\times10^{-8} &8.7\times10^{-4} &4.7\times10^{-5} &0.003 &2.9\times10^{-6}    \\\\
> > \textrm{v.s. DROCC} &9.4\times10^{-4} &0.004 &3.3\times10^{-4} &3.4\times10^{-6} &3.0\times10^{-5}  &3.1\times10^{-6} &4.4\times10^{-4}  &3.4\times10^{-7} &9.9\times10^{-6} &5.1\times10^{-4}   \\\\
> > \textrm{v.s. HRN}  &4.0\times10^{-9} &0.614 &1.2\times10^{-7} &0.013 & 5.9\times10^{-6} &3.0\times10^{-10} &1.5\times10^{-9} &5.4\times10^{-4} &2.2\times10^{-5} &1.7\times10^{-10}   \\\\
> > \hline
> > \end{matrix}
> > $$
> > Finally, it should be pointed out that, in the experiment of identifying anomaly from multi-class data mentioned in Section 4.3.3, all results of the competitors were obtained by running the released codes of the authors.
> >
> > **Q3: It's not clear if the results are statistically significantly different. A rigorous statistical analysis is desirable.**
> >
> > **Response:** Thanks for pointing out. To evaluate whether the results are statistically significant or not, we not only added the standard deviations but also added the results ($p$-value) of Student's t-test. Usually, the difference is said to be significant if the $p$-value is less than 0.05. The related content can be found in the revised version of Table 3 in the main text, as well as in the supplementary material. We show the $p$-value in t-test of our method against DSVDD, DROCC and HRN on Fashion-MNIST in the table above. We can see that the differences are statistically significant in all cases except the comparison with HRN on the ''Trouser'' and ''Dress'' classes.
> >
> > **Thank you again for your comments and evaluation. Please feel free to let us know if your have more questions.**

---

> ### Author Response · Authors · 2022-08-09
> **Have we addressed your concerns?**
>
> Hi Reviewer,
>
> As the author-reviewer discussion phase closes today, we'd like to know whether we have addressed your concerns or not. If yes, could you please modify the rating appropriately? If not, please let us know the issues and we will take advantages of the remaining hours to provide more explanations. Thanks a lot.
>
> Sincerely,
>
> Authors

---

### Meta-Review · Area_Chair_9xyo · 2022-08-24

**Recommendation:** Accept
**Confidence:** Less certain

**Metareview:**

This paper proposes an anomaly detection approach that learns perturbations directly from the normal training data. Although the reviewers are concerned about the novelty and depth of the approach, they appreciate the comprehensive experimental results. Therefore, I recommend acceptance for the paper.

**Award:**

No

---

### Decision · Program_Chairs · 2022-09-14

Accept